# Lipids and Lipid Mediators Associated with the Risk and Pathology of Ischemic Stroke

**DOI:** 10.3390/ijms21103618

**Published:** 2020-05-20

**Authors:** Anna Kloska, Marcelina Malinowska, Magdalena Gabig-Cimińska, Joanna Jakóbkiewicz-Banecka

**Affiliations:** 1Department of Medical Biology and Genetics, Faculty of Biology, University of Gdańsk, Wita Stwosza 59, 80-308 Gdańsk, Poland; anna.kloska@ug.edu.pl (A.K.); marcelina.malinowska@ug.edu.pl (M.M.); 2Laboratory of Molecular Biology, Institute of Biochemistry and Biophysics, Polish Academy of Sciences, Kładki 24, 80-822 Gdańsk, Poland

**Keywords:** eicosanoids, cholesterol, ischemic stroke, ischemia, lipoproteins, polyunsaturated fatty acids

## Abstract

Stroke is a severe neurological disorder in humans that results from an interruption of the blood supply to the brain. Worldwide, stoke affects over 100 million people each year and is the second largest contributor to disability. Dyslipidemia is a modifiable risk factor for stroke that is associated with an increased risk of the disease. Traditional and non-traditional lipid measures are proposed as biomarkers for the better detection of subclinical disease. In the central nervous system, lipids and lipid mediators are essential to sustain the normal brain tissue structure and function. Pathways leading to post-stroke brain deterioration include the metabolism of polyunsaturated fatty acids. A variety of lipid mediators are generated from fatty acids and these molecules may have either neuroprotective or neurodegenerative effects on the post-stroke brain tissue; therefore, they largely contribute to the outcome and recovery from stroke. In this review, we provide an overview of serum lipids associated with the risk of ischemic stroke. We also discuss the role of lipid mediators, with particular emphasis on eicosanoids, in the pathology of ischemic stroke. Finally, we summarize the latest research on potential targets in lipid metabolic pathways for ischemic stroke treatment and on the development of new stroke risk biomarkers for use in clinical practice.

## 1. Introduction

Stroke is a severe neurological disorder in humans that results from an interruption of the blood supply to the brain caused by a vascular occlusion (ischemic stroke, IS) or a blood vessel rupture or leakage (hemorrhagic stroke, HS). Within seconds, the insufficient blood supply leads to a strong oxygen-glucose deprivation (OGD) of the brain tissue, which initiates a cascade of pathophysiological response consequently leading to neuronal death and severe neurological deterioration. According to the Global Burden of Disease study, in 2017, stroke affected 104.2 million people worldwide; of them, 82.4 million were affected by ischemic stroke. Stroke is the second largest contributor to disability-adjusted life years in the world after ischemic heart disease, resulting in up to 50% of survivors being chronically disabled, and is the second most common cause of death [1,2]. The age-standardized global rate of new strokes reached 150.5 per 100,000 people in 2017 [3].

Primary and secondary prevention of stroke is crucial, considering that over 80% of the global stroke burden is attributable to a few risk factors that can be improved significantly. Many risk factors for stroke have been documented, including hypertension, current smoking, diabetes, abdominal obesity, poor diet, inactivity, excessive alcohol consumption, cardiac causes and stress/depression [4]. Dyslipidemia is a modifiable risk factor for stroke and is associated with a 1.8- to 2.6-times relative risk of stroke.

In the central nervous system, lipids and lipid mediators are essential to sustain the normal structure and function of brain tissue. Pathways leading to post-stroke brain deterioration include the metabolism of polyunsaturated fatty acids (PUFAs). The rapid and extensive release of PUFAs from cell membranes starts in the brain tissue immediately after the onset of ischemia. The lipids released are utilized in either enzymatic or non-enzymatic reactions, generating diverse classes of short-lived, lipid mediators, e.g., eicosanoids. These molecules have either neuroprotective or neurodegenerative effects on the post-stroke brain tissue; thus, they largely contribute to the outcome and recovery from stroke. Because the brain infarct area consists of two zones—the ischemic core, which is generally considered unsalvageable because of the greatest damage, and the surrounding area called the penumbra, where the blood flow during stroke is only partially reduced—the brain cells located in the penumbra may be rescued from degeneration by timely intervention. Numerous PUFA-derived lipid mediators contribute to the brain injury occurring after the ischemic stroke. 

In this review, we provide an overview of serum lipids associated with the risk of ischemic stroke. We also discuss the role of lipid mediators, with particular emphasis on eicosanoids, in the pathology of ischemic stroke. Finally, we summarize the latest research on potential targets in lipid metabolic pathways for ischemic stroke treatment and on the development of new stroke risk biomarkers for use in clinical practice.

## 2. Association of Serum Lipids with the Risk of Stroke

Dyslipidemia is conventionally considered to play an important role in the pathogenesis of stroke, primarily ischemic stroke. Traditional lipid parameters, represented by increased concentrations of total cholesterol (TC), triglycerides (TGs), low-density lipoprotein cholesterol (LDL-C), and decreased high-density lipoprotein cholesterol (HDL-C), have been identified as risk factors and predictors of cardiovascular disease, including stroke [5,6,7]. The assessment of lipid ratios such as TC/HDL-C, TG/HDL-C, and LDL-C/HDL-C is recognized as a better predictor of vascular risk compared to traditional lipid parameters [8]. In addition to standard lipid components testing, the analysis of composition, particle size, and density of lipids and lipoproteins [e.g., lipoprotein(a), [Lp(a)]], have been proposed as biomarkers for the better detection of subclinical diseases [9].

Lipid profile components and the risk of ischemic stroke are discussed in subsequent paragraphs and summarized in Figure 1.

### 2.1. Cholesterol and the Risk of Stroke

As one modifiable risk factor, TC has been shown by many studies to be associated with the risk of stroke [10]. Although the relationship between lipid levels and coronary heart disease (CHD) is well established, the results of observational studies investigating the relationship between lipid profile and stroke are less conclusive. Some data from Asian and American studies indicate that TC is not identified as being associated or shows only weak relationships with various stroke subtypes [11,12]. Until the beginning of the 21st century, research on this topic yielded inconsistent results [13,14,15] and led to conflicting views on the importance of circulating cholesterol in IS [5,16]. Part of the controversy is related to the methodological shortcomings of previous studies. Many did not investigate the final point of interest, namely the ischemic stroke incident, but focused rather on endpoints such as fatal stroke or a combination of IS and HS, which are clearly different pathophysiological units. Some studies lack data on cholesterol subfraction levels or rely on lipid profile measurements only after a stroke has occurred. Additionally, few studies are large enough to examine the dose–response pattern in each sex group, which is another important aspect of the nature of this disease [11].

In later years, a number of studies appeared that took into account the previous suggestions of scientists. Some reports have shown a positive association between TC and IS [17], whereas an inverse relationship between TC and HS was found in others [17,18]. In a meta-analysis, primarily in Europe and North America, low-density lipoprotein cholesterol is associated with ischemic stroke but not with hemorrhagic stroke [19]. In addition, elevated low-density lipoprotein cholesterol is shown to be one of the most important risk factors for coronary artery disease and stroke, while high-density lipoprotein cholesterol is protective [20,21,22]. A study by Tirschwell et al. shows that elevated cholesterol and reduced HDL-C levels are associated with an increased risk of IS [6]. The effect of hyperlipidemia may be different depending on the ischemic stroke subtype. A significant and positive relationship was found in the case of atherothrombotic infarction, but a negative association was observed in the case of cardioembolic infarction [23].

Too low values of lipid components may also be responsible for an increased risk of stroke. The risk of HS is higher when TC is lower than 120 mg/dL. LDL-C and TC seem to be associated with hemorrhagic stroke. In contrast, the risk of ischemic and hemorrhagic stroke may be higher when HDL-C is lower than 50 mg/dL [24].

### 2.2. Hypertriglyceridemia and Ischemic Stroke

Few reports to date have shown the role of triglycerides in acute stroke and recovery after stroke. The studies that have evaluated this association report diverse associations [7,25,26]. According to a study published in 2012 in the *Stroke* journal, the strongest predictor of stroke risk in women is TC, the most-overlooked lipid in the cholesterol profile. The authors found that people with the highest TC levels are 56% more likely to have ischemic stroke than those with the lowest levels [27]. Triglyceride levels and their role in ischemic stroke have already been the subject of intensive research [28,29]. Large-scale epidemiological studies also detected a relationship between high non-fasting triglyceride levels and the risk of ischemic stroke [30,31], while results in smaller cohorts remain inconclusive [29]. Weir et al. suggested the opposite results and showed that low TG level, not low TC concentration, independently predicts poor outcome after acute stroke [32]. The mechanisms by which triglycerides affect ischemic stroke are still unexplained [33]. Triglyceride-rich lipoproteins and high TC levels can have a direct atherogenic effect and appear to be an indicator of atherosclerotic and prothrombotic changes. Elevated TC levels are associated with abnormalities in the coagulation cascade and fibrinolysis that is associated with ischemic stroke [34]. The effects of triglycerides are likely to be multifactorial.

### 2.3. Non-Traditional Lipid Profiles as Stroke Predictors

Human lipid profiles are currently actively studied, including not only traditional but also non-traditional lipid profiles, primarily as independent predictors of cardiovascular disease (CVD) [35,36]. Some studies show that low-density lipoprotein cholesterol, non-high-density lipoprotein cholesterol (non-HDL-C) and the TC/HDL-C ratio are significant predictors of CVD [37,38]. The relationship between LDL-C/HDL-C lipid profiles is found to be a more useful indicator of CVD risk than single, isolated lipid values [39]. In addition, some reports show that non-HDL-C is a better indicator of the development of CVD than LDL-C [36]. Lipid profiles as the main indicator of stroke prevention are still subject to significant uncertainty, in contrast to the clear results for CVD. However, several studies attempted to assess the role of traditional and non-traditional lipid indices in predicting stroke risk (Figure 1). Among the various non-traditional lipid variables, elevated baseline TC/HDL-C ratio and TC/HDL-C ratio increases future vascular risk after stroke, but only elevated TC/HDL-C ratio is associated with stroke recurrence risk [8]. Other studies estimated that the recurrence of cerebral ischemia increases with age and the increased composition of non-traditional lipid variables values: TC/HDL-C and LDL-C/HDL-C [40]. Zheng et al. showed that LDL-C, non-HDL-C and LDL-C/HDL-C are associated with the future all stroke status, and TC, LDL-C, non-HDL-C, TC/HDL-C and LDL-C/HDL-C are associated with the future ischemic stroke state [41]. Other studies show a positive relationship between cholesterol levels and the risk of IS in men, while an inverse trend between TC and the risk of hemorrhagic stroke is observed in the female group. A positive relationship was found between TC/HDL-C ratio and risk of ischemic stroke in both sexes; however, these links are not as clear after adjusting for body mass index, blood pressure and history of diabetes [17]. Other authors also studied the impact of lipid profiles separately in men and women. TC/HDL-C ratio is mainly associated with ischemic stroke and total stroke in men, while TG is more important in predicting ischemic and total stroke in women. The authors of this study suggest that these two lipid indexes have the most important prognostic value for identifying high risk participants predisposed to stroke for each sex separately, and there may be potential goals for stroke prevention [42]. Some researchers recommend that the level of non-traditional lipid profiles should be considered in the daily treatment of ischemic stroke for the first prevention in clinical practice [8].

### 2.4. Number, Size and Composition of Lipoprotein Particle

Standard measurements of circulating lipids lack the ability to distinguish between the size, density or concentration and composition of lipoproteins that may be important in assessing CVD risk [43]. Therefore, in addition to the standard lipid components tested, several biomarkers of lipids and lipoproteins are proposed as potential risk factors for the better detection of subclinical diseases [44]. Conventional HDL-C measures include the sum of cholesterol carried in HDL particles, but ignore their composition, particle size, and subclass concentration. Additionally, TG may show divergent relationships with vascular disease when transported in different lipoprotein molecules [9]. Above all, the interest focuses on lipoprotein parameters such as the number and size of LDL and HDL particles, and the number of intermediate-density lipoprotein (IDL) particles and lipoprotein(a) [Lp(a)]. One of the most commonly used methods for measuring the size and concentration of lipoprotein particles is nuclear magnetic resonance (NMR) spectroscopy. This technique simultaneously determines the average size (in nanometers) and concentration (in mol/L) of lipoprotein particles [45]. Holmes et al. assessed the relationship between metabolic markers and the risk of three cardiovascular diseases, including ischemic stroke. The study shows that the subclasses of lipoproteins and their lipid components are associated with stroke risk. Cholesterol and triglycerides in apolipoprotein B-containing lipoproteins (very low-density lipoprotein [VLDL], intermediate-density lipoprotein [IDL] and low-density lipoproteins [LDL]) are positively associated with the risk of stroke. In contrast, cholesterol in large and medium high-density lipoprotein particles inversely associates with the ischemic stroke risk, while triglycerides in HDL particles positively associate with the risk of this disease. VLDL particle concentrations are at least as strongly associated with ischemic stroke as LDL particles. In addition, TGs are more consistently associated with IS across the entire lipoprotein subfraction spectrum than cholesterol [9].

### 2.5. Elevated Lipoprotein(a) and the Risk of Ischemic Stroke

Lipoprotein(a) is an LDL particle with an added apolipoprotein(a). The link between Lp(a) and stroke has been questioned by some researchers. Several studies show no association between Lp(a) levels and the risk of stroke. Hachinski et al. did not notice a significant difference in Lp(a) levels among the patient and control groups [46]. Similarly, Glader et al. found no association between baseline plasma Lp(a) values and future ischemic cerebral infarction [47]. In a prospective study in Finland, no relationship was found between the initial Lp(a) plasma levels and the future risk of total (all types of stroke) or thromboembolic stroke among those participating in the study [48]. More recent meta-analyses summarize the existing evidence for Lp(a) and stroke from both control and prospective studies and show a significant and independent association of elevated Lp(a) with an increased risk of ischemic stroke [49,50,51]. The largest meta-analysis carried out by the Emerging Risk Factor Collaboration analyzed the data from 13 prospective studies and confirmed that elevated Lp(a) is an independent risk factor for ischemic stroke [49]. As many previously published reports investigating the risk of stroke in patients with elevated Lp(a) levels did not show the underlying cause of the stroke event; differences in the etiology of stroke between cohorts may explain some of the inconsistent results reported in the literature. Data suggest that Lp(a) primarily increases the risk of large-artery atherosclerosis stroke [52]. According to the guidelines of the European Society of Atherosclerosis, it is recommended to measure Lp(a) in patients with a medium or high risk of cardiovascular disease, considering levels lower than 50 mg/dL [53]. Studies with larger cohorts are needed to see whether higher cut-off values than conventional and/or interactions with other risk factors are necessary to establish the role of Lp(a) as a risk factor for ischemic stroke. Therefore, a more detailed study of the relationship between individual subclasses of lipoprotein particles and lipid-associated characteristics with the risk of CVD and stroke subtypes may be important and informative.

### 2.6. Polyunsaturated Fatty Acids and Risk of Ischemic Stroke

The n-3 polyunsaturated fatty acids (n-3 PUFAs) are a class of essential unsaturated fatty acids necessary for proper biological activity and function in living organisms. The n-3 PUFAs are poorly synthesized in the human body and have to be orally supplemented. Fish, such as mackerel, salmon, tuna, sardines, herring, and halibut, are a major source of n-3 PUFAs in the human diet, and they contain docosahexaenoic acid (DHA), docosapentaenoic acid (DPA), and eicosapentaenoic acid (EPA) [54]. The n-3 PUFAs have potent anti-inflammatory activity, reduce platelet aggregation, stabilize atherosclerotic plaques, and reduce major cardiovascular risk factors, including hypertension and hyperlipidemia [55]. They can act as an antioxidant in reducing cerebral lipid peroxides and play a role in regulating oxidative stress by increasing the oxidative burden and improving antioxidant defense capabilities [56]. In addition, n-3 PUFAs trigger other responses, such as neurogenesis and revascularization in stroke, which could be used in the development of acute-phase ischemic stroke therapy [54]. Moreover, because ischemic stroke is a heterogeneous disorder with different pathophysiological pathways and separate etiological subtypes, various mechanisms of PUFA action should be considered. DHA plays a greater role in reducing the risk of atherothrombotic stroke by reducing endothelial dysfunction and atherosclerosis [57], while EPA and DPA have a greater impact on the risk of cardioembolic stroke because of their effect on clotting and atrial fibrillation [58]. The second key component of a heart-healthy dietary pattern is a high content of n-6 PUFAs, obtained mainly from vegetable oil, nuts, and seeds. Dietary n-6 PUFAs primarily include linoleic acid (LA) and arachidonic acid (AA). The n-6 PUFAs have not generally been associated with stroke risk [59,60].

The Japan EPA Lipid Intervention Study (JELIS) has shown that treatment with highly purified EPA and low-dose statins significantly reduces the incidence of coronary artery diseases and stroke compared to statin therapy alone, without altering the reduction in low-density lipoprotein cholesterol levels [61]. Nishizaki et al. reported that ratios of serum n-3 PUFAs to n-6 PUFAs, such as EPA/AA and DHA/AA ratios, could be useful markers to determine the incidence of coronary events, peripheral artery diseases, and early neurological deterioration after acute ischemic stroke [62]. Thies et al. examined the effect of fish oil administration on plaque regression and found that giving fish oil to patients resulted not only in plaque regression but also an increases in EPA and DHA within the plaque and a decrease in macrophage counts [63]. In addition, Ajami et al. reported that DHA+EPA provided neuroprotection against ischemic brain injury by increasing the levels of antiapoptotic proteins, such as Bcl-2 and Bcl-xL, thereby suppressing the inflammatory response [64].

Attention has also been focused on assessing the relationship between PUFA levels and early neurological deterioration (END) in acute-phase ischemic stroke [65]. END occurs in approximately one-third of patients in the acute phase of ischemic stroke and is associated with neurological and functional decline. It also strongly correlates with poor functional outcome and usually leads to a significant increase in mortality rate [66]. According to a recommended definition, END occurs when there is an increase in the total National Institutes of Health Stroke Scale score of ≥2 points within 72 h 1–3 times a day after admission [67]. Suda et al. revealed that END is negatively associated with the EPA/AA, DHA/AA, and EPA+DHA/AA ratios. The study shows that a low serum n-3 PUFA/n-6 PUFA ratio might be an indication of possible END in patients with acute ischemic stroke, as demonstrated in the population of Japanese stroke patients [65].

### 2.7. Fatty Acids and Cardioembolic Stroke

Elevated fatty acid (FA) levels are associated with several risk factors for atherosclerosis, including abdominal obesity [68], arterial hypertension [69], and insulin resistance [70], as well as coronary artery disease (CAD) [71], arrhythmia [72], and atrial fibrillation [73]. Because an association with both atherosclerosis and arrhythmia has been reported, FAs may correlate with ischemic stroke. However, the effect of FAs on ischemic stroke is poorly understood. Because ischemic stroke is a heterogeneous disorder with a variety of pathophysiological pathways, including atherothrombosis and cardioembolism, the etiological subtypes of ischemic stroke should be analyzed separately [74]. Almost all long-term correlation studies of FAs and stroke risk estimate the level of FAs obtained with food on the basis of self-reported questionnaires. Unfortunately, this is unclear for individual FAs that are not well separated by dietary questionnaire data. For a more detailed assessment, Saber et al. measured the levels of circulating n-3 PUFA phospholipids and examined their association with ischemic stroke incidence, including atherosclerotic and cardioembolic stroke subtypes [75]. Patients evaluated for phospholipid levels were recruited for three separate prospective cohort studies in the US: the Cardiovascular Health Study (CHS), Nurses’ Health Study (NHS), and Health Professionals Follow-Up Study (HPFS) [75]. The authors of these studies show that, among ischemic stroke subtypes, DHA is inversely associated with atherothrombotic stroke and DPA is associated with cardioembolic stroke. These relationships remain significant after including demographic, lifestyle, and vascular risk factors. By comparison, EPA is not associated with total ischemic, atherothrombotic, or cardioembolic stroke. The authors confirmed the hypothesis that individual FAs in serum have various associations with ischemic, atherothrombotic, and cardioembolic stroke [75]. Earlier studies have shown that elevated FA levels are associated with cardioembolic (CE) stroke, but this association was not seen in non-CE stroke. Atrial fibrillation may potentially act as an intermediary between FAs and stroke caused by cardioembolism [73]. Another study also found that an elevated FA concentration may serve as a marker of stroke caused by cardioembolism. In addition, the assessment of FA concentration can predict the stroke recurrence following a CE stroke [76]. In patients with acute stroke, significantly elevated FA levels are observed in groups with a higher risk of cardioembolism. These results suggest that enhanced thrombogenicity may be the main mechanism explaining elevated FA levels in patients with cardioembolic stroke [77].

### 2.8. The Role of Fish-Derived Fatty Acids in Stroke

For many years, intensive research has been carried out to assess the biological effects of consuming fish-derived PUFAs [78]. These studies confirm the view that n-3 PUFAs can affect several cellular processes known to be important in the development of cardiovascular disease, stroke, and protective effects [79,80]. Most studies have used doses of fish oils exceeding what is usually found in the diet. Surprisingly, significant vascular benefits are observed even with modest fish consumption [81]. Long-term studies have shown that increased intake of n-3 PUFAs, in particular EPA and DHA, can have a beneficial effect on serum lipids [82], platelet aggregation [83], and bleeding time [84] and, thus, may lead to a reduced risk of atherosclerosis and thrombotic complications [63]. Prolonged fish consumption leads to increased incorporation of n-3 PUFAs into plasma lipids, erythrocytes, and platelets [85]. After supplementation with fish oil, elevated EPA and DHA content in plasma lipids, platelets, and erythrocyte membranes are observed with a simultaneous decrease in AA content [86]. In order to compare the effects of n-3 PUFA consumption, the following four fish-derived sources of n-3 PUFA were used: three rich sources (raw fatty fish [smoked salmon], cooked fatty fish [salmon fillet], or fish oil [cod liver oil]) and one poor source (fish low in n-3 PUFA [cod fillet]). Therefore, the following blood parameters were assessed: blood lipid composition and functional properties of blood cells, as measured by the potential of lipopolysaccharide (LPS) to generate activation products in whole blood [87]. Elvevoll et al. did not notice any significant differences between the effects of eating cooked fish compared to raw fish (smoked salmon). They found that the intake of fatty fish is more effective in increasing EPA and DHA than supplementing with fish oil and is more likely to have a beneficial effects on HDL cholesterol and whole blood activation reactions [87]. However, another group of researchers showed that the action of DHA-rich oil (without EPA) had a comparable hypotriglyceridemic effect as a fish diet and fish oil supplementation. Moreover, a fish diet and fish oil supplementation increased the proportion of n-3 PUFA in plasma lipids, platelets, and erythrocyte membranes [88].

### 2.9. Lipid-Lowering Therapy for Prevention of Ischemic Stroke

Over the past two decades, compelling evidence from clinical trials revealed the importance of low-density lipoprotein cholesterol-lowering therapies in reducing cardiovascular and stroke morbidity and mortality [89,90]. At present, there is a more than four-fold increase in the use of cholesterol-lowering agents in our population compared to in 2000. There is strong evidence for the role of statins in stroke prevention and association with an approximately their 20% risk reduction [91], in particular a decreased risk of ischemic stroke [92]. Moreover, more aggressive statin treatment improves the long-term functional outcome of patients discharged after an acute ischemic stroke more than less aggressive treatment [93]. One of the most commonly used statins to lower high cholesterol is atorvastatin. It is a common statin used to investigate the efficacy of statin therapy, especially high-intensity statin therapy in patients with ischemic stroke. Preliminary studies have shown that in patients with recent stroke or transient ischemic attack and without coronary heart disease, 80 mg atorvastatin daily reduces the overall incidence of stroke and cardiovascular events, despite a slight increase in the incidence of hemorrhagic stroke [94]. Many studies have confirmed the effect of atorvastatin at lowering lipids and decreasing the number and frequency of vascular incidents, as well as its clinical efficacy at reducing the burden of disease after stroke or transient ischemic attack [95,96]. Atorvastatin improves endothelial function, enhances the stability of atherosclerotic plaque, and inhibits inflammatory and thrombogenic responses in arterial walls [97]. In the latest Stroke Prevention by Aggressive Reduction in Cholesterol Levels (SPARCLs) cohort trial, atorvastatin was compared with a placebo in patients with recent stroke or transient ischemic attack. Atorvastatin was found to reduce the first occurrence of stroke and the first occurrence of composite vascular events [98].

During recent years, new drug classes proved their efficacy and safety in lowering cholesterol and preventing cardiovascular incidents in randomized controlled studies. PCSK9 (proprotein convertase subtilisin/kexin type 9) inhibitors increase the number of available LDL receptors on the surface of hepatocytes, which leads to a higher cleavage of LDL-C from the circulation. The mechanism of action of PCSK9 inhibitors involves halting the metabolic breakdown of LDL-R, resulting in increased LDL-C clearance. The two currently available antibodies (Alirocumab and Evolocumab) against PCSK9 are fully human IgG subtypes that bind with an approximate 1:1 stoichiometry to circulating PCSK9 and prohibit its binding to the LDL-R. The mean percentage change in LDL-C levels decreases by 50% or more from baseline in patients receiving PCSK9 inhibitors [99]. Several non-antibody therapies have also been developed to inhibit PCSK9 function. Gene silencing or editing technologies were used, such as antisense oligonucleotides [100], small interfering RNAs [101], small-molecule inhibitors [102], mimetic peptides [103], adnectins [104], and vaccinations [105].

According to the American Heart Association and American Stroke Association guidelines for stroke prevention, which recognize triglyceride as a risk factor for stroke, fibric acid derivatives can be considered in patients with hypertriglyceridemia [106]. Fibric acid derivatives (e.g., gemfibrozil, fenofibrate, and bezafibrate) lower triglyceride levels and increase HDL cholesterol. The Veterans Administration HDL Intervention Trial of men with low HDL-C and associated coronary artery disease showed that gemfibrozil reduces the risk of all strokes, mainly ischemic strokes [107].

The issue of elevated Lp(a) level therapy is slightly different. Lp(a) levels are essentially unresponsive to traditional lipid-lowering drugs such as statins or fibrates. Some lipid-lowering agents that are not specific for Lp(a) reduce Lp(a) levels (e.g., niacin, PCSK9 inhibitors, and CETP inhibitors) [108]; however, to date, no randomized controlled trial has demonstrated that the lowering of Lp(a) leads to decreased risk of cardiovascular disease. Recent findings show that antisense oligonucleotides can be used to inactivate genes involved in the pathogenesis of vascular diseases. To lower Lp(a), synthetic oligonucleotides have been developed. In clinical trials, Mipomersen (Genzyme/ISIS Pharmaceuticals), an antisense oligonucleotide targeted to apolipoprotein B, reduced Lp(a) by 21% to 39% [109]. AKCEA-APO(a)-LRx (Akcea Therapeutics/Ionis Pharmaceuticals) is the latest antisense oligonucleotide targeted to apolipoprotein(a). In a phase II trial, the specific antisense oligonucleotide resulted in a dose-dependent reduction of 66% to 92% in blood circulating Lp(a) and is expected to enter into a phase III study in 2020.

## 3. Lipids of the Brain during Ischemic Stroke

The brain has the second highest lipid content among the organs of the human body, accounting for about 50% of its dry weight [110]. Lipids that are essential for the central nervous system are classified into five major subcategories: fatty acids, triglycerides, phospholipids, sterol lipids, and sphingolipids. They serve as the structural components of biological membranes, act as messengers in cellular signaling pathways and contribute to the energy supply [111]. For example, about 20% of the brain total energy requirements comes from the oxidation of fatty acids, which takes place in astrocytes [112]. 

Brain tissue is characterized by cellular heterogeneity; thus, the fatty acid composition varies between different cell types. In the ischemic brain, the levels of different lipids change compared to the control conditions [113]. It is even possible to distinguish the ischemic core area from the penumbra according to the post-stroke lipid profiles [114]. Generally, brain tissue is characterized by a high proportion of polyunsaturated fatty acids: arachidonic acid (AA; omega-6; C20:4ω6), eicosapentaenoic acid (EPA; omega-3; C20:5ω3), and docosahexaenoic acid (DHA; omega-3; C22:6ω3). Arachidonic acid and docosahexaenoic acid make up ∼20% of fatty acids in the mammalian brain [115]. All of these lipids are located in cellular membranes and are essential for the normal structure and function of the central nervous system [110]. However, cerebral ischemia initiates a cascade of events that stimulates the release of free fatty acids from the membrane. In particular, there is a rapid accumulation of arachidonic acid, docosahexaenoic acid, diacylglycerol, and platelet-activating factor (PAF; 1-Oalkyl-2-acyl-sn-3-phosphocholine).

The released arachidonic acid is highly reactive and prone to downstream enzymatic reactions that generate different classes of eicosanoids in the brain area affected by the ischemia (Figure 2). These molecules play a major role in cerebral vasoconstriction, edema, neurotoxicity and neuroprotection that occur after ischemia and reperfusion in the brain following stroke episodes. The cytosolic phospholipase A_2_ alpha (cPLA_2α_), an enzyme responsible for arachidonic acid release from membrane phospholipids, plays a key role in post-ischemic brain pathology. The down-regulation of cytosolic phospholipase A_2_ alpha expression [116] or blockage of its activity with specific antibodies [117] attenuates ischemic brain damage in the mouse model of cerebral ischemia-reperfusion injury.

The DHA released during the ischemic event is converted enzymatically into lipid messengers. One of them is the neuroprotectin D1 (NPD1), a potent cell-protective, pro-survival and anti-inflammatory lipid mediator in experimental stroke models [118,119]. This experimental post-stroke DHA treatment improves the neurobehavioral recovery and reduces brain infarct volume. This research shows that the additional DHA enhances the synthesis of NPD1 in the penumbra, revealing a highly desirable neuroprotective effect [118].

The excessive accumulation of platelet-activating factor in the ischemic brain contributes to excitotoxicity, Ca^2+^ uptake, and mitochondrial dysfunction, leading to neuronal death. However, the disruption or inhibition of PAF-receptor has a neuroprotective effect [120,121].

## 4. Eicosanoids in Ischemic Stroke

Eicosanoids belong to oxylipins, a family of oxidized forms of polyunsaturated fatty acids. They are produced by enzymatic reactions catalyzed by three different enzymes—cyclooxygenases, lipoxygenases and cytochrome P450 monooxygenases. This diverse group of compounds includes prostanoids (i.e., prostaglandins, prostacyclin, and thromboxanes), leukotrienes, lipoxins, hydroxyeicosatetraenoic acids (HETEs), epoxyeicosatrienoic acids (EETs), resolvins, and eoxins. In general, eicosanoids are biologically active compounds involved in the regulation of many physiological functions, but they are also related to pathological processes, including ischemic stroke.

### 4.1. Biosynthesis and Physiological Role of Eicosanoids

Eicosanoids are synthesized from arachidonic acid, a precursor, 20-carbon polyunsaturated fatty acid released from cell membranes by phospholipase A_2_ (PLA_2_). Free arachidonic acid, as a highly reactive compound is prone to further oxidation and is readily converted into eicosanoids in their downstream enzymatic pathways (Figure 2). Cyclooxygenases (COX-1 and COX-2) convert the free arachidonic acid into the unstable prostaglandin H_2_ (PGH_2_), which is next converted by downstream prostanoid synthases into prostaglandins D_2_, E_2_, F_2_, and I_2_ (PGD_2_, PGE_2_, PGF_2_ and PGI_2_, respectively) or thromboxane A_2_ (TXA_2_) [122]. Arachidonic acid is also converted by lipoxygenases (LOX enzymes) to produce leukotrienes, lipoxins and HETEs, while cytochrome P450 converts the arachidonic acid into 20-HETE and EETs molecules.

Eicosanoids are known for their various and often contradictory roles in the human body. On the one hand, they regulate many physiological functions in the cardiovascular, gastrointestinal, urogenital, and nervous system. They play critical roles in immunity, acting as pro- and anti-inflammatory agents, regulating fever, and triggering platelet aggregation, blood clotting, muscle contraction, vasoconstriction, and vasodilation. On the other hand, eicosanoids play an important role in cardiovascular diseases and stroke, renal diseases, rheumatoid arthritis, Alzheimer’s disease, cancer, and even infectious diseases [123,124].

### 4.2. The Role of Eicosanoids in Ischemic Stroke Pathology

The role of eicosanoids in stroke is reported to be both beneficial and detrimental. The levels of many eicosanoids are altered following the ischemic insult and often stay elevated during the recovery phase, thus playing an important role in either brain tissue injury or protection. The balance between neuroprotection and neuronal cell death caused by the post-stroke disruption of eicosanoid homeostasis is detrimental for the size of the disease pathology and the extent of stroke patient recovery. In the following sections, principal findings of the current investigations on the role of different classes of eicosanoids in ischemic stroke pathology as well as possible targets for therapeutic interventions will be reviewed.

#### 4.2.1. Prostanoids in Ischemic Stroke

Prostanoids group arachidonic-acid derivatives which are generated by cyclooxygenase enzymes. These include prostaglandins PGE_2_, PGD_2_, PGF_2α_, PGI_2_, and thromboxane A_2_ (Figure 2). Two distinct isoforms of cyclooxygenase are involved in the prostanoid biosynthetic process: COX-1, which is found in the kidney, stomach and platelets, and COX-2, located in macrophages, leukocytes and fibroblasts. In the central nervous system, COX-1 is constitutively expressed in neurons, astrocytes, and microglial cells, while COX-2 is up-regulated under pathological conditions [125].

Special interest in the pathogenesis of ischemic stroke concerns the role of two prostanoids: prostacyclin, also known as prostaglandin I_2_ (PGI_2_), and thromboxane A_2_ (TXA_2_). Prostacyclin is a potent vasodilator and inhibitor of platelet aggregation. In contrast, thromboxane A_2_ is a strong vasoconstrictor and inductor of platelet aggregation [126]. The over-production of TXA_2_ is one of the key factors causing thrombosis, stroke, and heart disease. PGI_2_ is the primary arachidonic acid metabolite in vascular walls; due to its contrasting biological activity to TXA_2_, PGI_2_ represents the most potent endogenous vascular protector, acting as an inhibitor of platelet aggregation and a strong vasodilator on vascular beds [127]. PGI_2_ acts mostly as an immune-inhibitory molecule through multiple cell types such as dendritic cells, macrophages, and T-cells. The decreased biosynthesis of anti-thrombotic PGI_2_ alongside with the excessive production of pro-inflammatory PGE_2_ and pro-thrombotic TXA_2_ increases the risk of stroke and heart attack in gene-knockout models [128]. Another study using the transgenic mouse model revealed that the redirection of arachidonic acid conversion toward favoring PGI_2_ production over TXA_2_ and PGE_2_ clearly contributes to resistance to induced ischemic stroke [128]. This experimental attempt uses genetic engineering to link the COX-1 enzyme, which generates the unstable prostaglandin H_2_ (PGH_2_) from arachidonic acid, with the PGIS enzyme that generates PGI_2_ from PGH_2_ intermediates. Since PGH_2_ serves as a substrate for the production of all of the downstream prostanoids (including PGI_2_, PGE_2_) and thromboxane A_2_, this approach enables the regulation of prostanoid biosynthesis in favor of PGI_2_ rather than TXA_2_ and PGE_2_ in cells by channeling the PGH_2_ intermediate to the PGIS biosynthetic pathway [128].

The post-stroke accumulation of PGE_2_ is also accompanied by a marked induction of the prostanoid EP2 receptor expression in the ischemic hemisphere, particularly in neurons of the penumbra [129]. The EP2 signaling pathway contributes to ischemic injury but its inhibition efficiently protects against the inflammatory neurodegeneration observed in post-ischemic brain tissue. In addition, the other EP1 prostanoid receptor is also up-regulated following ischemic stroke and its expression is not only detected in neurons but also in the endothelial cells [130]. The PGE_2_ action, mediated by the EP1 receptor, leads to the disruption of the blood–brain barrier which, in turn, significantly contributes to progressive neuronal death in the stroke penumbral region. Additionally, in this case, inhibition of the EP1 receptor reduces the infarct volume, blood–brain barrier disruption and permeability, neutrophil infiltration and hemorrhagic transformation in the animal stroke model.

#### 4.2.2. Leukotrienes in Ischemic Stroke

Leukotrienes (LTs) are synthesized form arachidonic acid by the 5-lipoxygenase (5-LOX) enzyme; this process also requires the presence of 5-LOX-activating protein (FLAP). The pathway generates two groups of leukotrienes: dihydroxy acid leukotriene B_4_ (LTB_4_) and cysteinyl-leukotrienes (i.e., LTC_4_, LTD_4_, and LTE_4_) (Figure 2). Leukotrienes are thought to be involved in atherosclerosis and plaque formation, a process leading to cardio- and cerebrovascular pathology that eventually leads to occlusion resulting in stroke. However, as leukotriene levels increase after ischemia, they are also considered the factors responsible for the post-stroke pathology.

Numerous studies pinpoint leukotriene B_4_ (LTB_4_) as the factor responsible for pathological events occurring during the post-ischemic state. Both the expression of 5-lipoxygenase and the levels of leukotrienes increase following focal cerebral ischemia in rats and persist for days after reperfusion in the ischemic core and the boundary zones [131]. The increased and sustained plasma level of leukotriene B_4_ is associated with the poor functional recovery of patients with acute middle cerebral artery infarction [132]. However, studies using animal or cell culture models consistently show that the regulation of 5-lipoxygenase expression results in a neuroprotective effect in cases of ischemia [133,134,135]. An evident post-ischemic brain injury is also mediated by the cysteinyl-leukotrienes but is associated with neuronal damage and astrogliosis [136].

#### 4.2.3. Lipoxin A_4_ in Ischemic Stroke

Lipoxins (LXs) are a class of eicosanoids that exert anti-inflammatory and pro-resolving activities, generally reducing tissue injury and chronic inflammation. The synthesis of LXs from arachidonic acid involves three major lipoxygenases: 5-LOX, 15-LOX, and 12-LOX (Figure 2). Additionally, LXs may be synthesized in an aspirin-triggered pathway [137].

One representative of this group of molecules, lipoxin A_4_ (LXA_4_), has been shown to be very effective in neuroprotection after brain ischemia-reperfusion injury. The experimental treatment with LXA_4_ effectively reduces infarct volume and brain edema, improving the neurological outcome, as shown for rats after middle cerebral artery occlusion [138,139]. The authors of the studies suggest that the effect obtained with the treatment may be generated by the inhibition of 5-lipoxygenase translocation triggered by LXA_4_ and the resulting reduced biosynthesis of pro-inflammatory leukotrienes [138]. 

The protective effect of lipoxin A_4_ also applies to astrocytes. The treatment of these cells with LXA_4_ protects against cell damage and attenuates the production of reactive oxygen species under oxygen-glucose deprivation and re-oxygenation conditions [140]. Up-regulation of the nuclear factor erythroid 2-related factor 2 (Nrf2) signaling pathway links lipoxin A_4_ with reduction of oxidative stress that most likely underlies its protective effect.

#### 4.2.4. Hydroxyeicosatetraenoic Acids in Ischemic Stroke

Hydroxyeicosatetraenoic acids (HETEs) are formed from arachidonic acid in three different metabolic pathways [141]. The predominant pathway involves lipoxygenases (5-LOX, 12-LOX, and 15-LOX enzymes) that form a variety of HETE molecules, including 5-HETE, 12-HETE, and 15-HETE (Figure 2). Small amounts of 11-HETE and 15-HETE can be generated by the cyclooxygenases COX-1 and COX-2. Some HETEs, with the 20-HETE molecule at the forefront, are generated by cytochrome P450 hydroxylases. The analysis of HETEs composition and quantity in rat brains exposed to ischemia reveals that the amount of HETE molecules tends to increase with time following the occlusion [142]. After 72 h, almost all lipoxygenase-generated HETEs are significantly increased in the affected brain tissue.

Animal studies show that hypoxia up-regulates the expression of 15-lipoxygenase in the brain artery endothelium after stroke and leads to the increased production of 15-HETE [143]. 15-HETE promotes angiogenesis and neuronal recovery that protects against ischemic brain infarction and improves neurological function in a mouse model of focal ischemia. It also stimulates the proliferation and migration of brain microvascular endothelial cells with the PI3K/Akt signaling pathway involved. Another study shows that 15-HETE boosts the angiogenesis in mouse ischemic brain by up-regulation of the vascular endothelial growth factor (VEGF) [144].

Contradictory results are obtained on the role of the 20-hydroxyeicosatetraenoic acid (20-HETE) in stroke pathology as it is either described as detrimental or beneficial. 20-HETE is produced by cytochrome P450 members, mainly CYP4A and CYP4F. The role of 20-HETE in ischemic brain pathology depends on whether it is determined in the early or late post-stroke phase. On the one hand, elevated 20-HETE levels that are detected in the plasma of ischemic stroke patients associate with greater lesion size and neurological impairment, reduced cognitive functions and poorer functional independence in daily living [145]. Some authors propose that this particular eicosanoid may serve as a valuable predictor of neurological deterioration prognosis in acute minor ischemic stroke as their findings indicate the association of high 20-HETE plasma levels and neurological deterioration and poorer prognosis in patients [146]. The increased production of 20-HETE is also observed in animal ischemia-reperfusion models [147,148]. For example, the expression of 20-HETE synthase increases in the plasma and brain of animals after experimental cerebral ischemia and clearly contributes to oxidative stress and endothelial dysfunction [148].

On the other hand, the increase in CYP4A expression and 20-HETE production following oxygen-glucose deprivation noted in astrocytes promotes angiogenesis via the induction of endothelial cell proliferation, tube formation and migration in a later stage of stroke [149]. The specific cross-talk discovered between astrocytes and endothelial cells is mediated by 20-HETE and involves HIF-1α/VEGF and JNK signaling pathways. The beneficial effects of 20-HETE on recovery from stroke are confirmed—the inhibition of CYP4A, an enzyme involved in 20-HETE synthesis, resulted in diminished peri-infarct angiogenesis and worsened neurological deficits in mice. The authors highlight that, although 20-HETE induces neuronal and vascular injury in the early post-stroke phase, in the later post-stroke phase, this mediator is necessary for neurovascular repair and remodeling to obtain functional recovery after stroke.

#### 4.2.5. Epoxyeicosatrienoic Acids in Ischemic Stroke

The synthesis of epoxyeicosatrienoic acids (EETs) from arachidonic acid is catalyzed by cytochrome P450 epoxygenases from the CYP2C and CYP2J families. Four biologically active EETs belong to this group of eicosanoids: 5,6-EET, 8,9-EET, 11,12-EET, and 14,15-EET (Figure 2). These molecules are metabolized to less bioactive dihydroxyeicosatrienoic acids (DHETs) by the soluble epoxide hydrolase (sEH).

EETs exert strong neuroprotective effects in the case of cerebral ischemia, especially with regard to components of the neurovascular unit. While oxygen-glucose deprivation decreases the viability of cerebral smooth muscle cells, different types of EETs prevent these events during in vitro experiments [150]. 14,15-EET shows the strongest anti-apoptotic effect under these conditions [150,151]. Two pathways are linked to the protective effects of EETs: PI3K/Akt pathway and ATP-sensitive potassium channels contribute to the EETs-protective effects on cerebral microvascular smooth muscle cells [151], while the JNK/c-Jun and mTOR signaling pathway contributes to the anti-apoptotic effect of 14,15-EET under conditions of oxygen-glucose deprivation [150].

### 4.3. Eicosanoids as the Target for Ischemic Stroke Treatment

The emphasis on the role of eicosanoids in the pathology of ischemic stroke has opened up new perspectives for the development of treatments based on modifications within the biosynthetic pathways of these lipid mediators.

The modulation of prostanoid biosynthesis with cyclooxygenase inhibitors has been proposed as one potential treatment strategy. To date, aspirin is the only antiplatelet agent that is used effectively in the early treatment of acute ischemic stroke. Aspirin irreversibly inhibits COX activity in platelets and prevents the conversion of AA to thromboxane A_2_. The decline in the risk of mortality and morbidity when aspirin is initiated within 48 h of acute ischemic stroke is small though significant [152]. Targeting prostanoid biosynthesis with COX-2 inhibitors, such as nonsteroidal anti-inflammatory drugs (NSAIDs), the classical pain medications, or the novel COX-2 selective inhibitors (e.g., celecoxib or rofecoxib), has been shown to reduce edema, neuroinflammation, and infarct size in rodent stroke models [153]. Although administration of these drugs reveals a neuroprotective effect in stroke models, adverse effects, including an increased risk of stroke, occur following long-term usage. A recent increase in the number of heart attacks related the use of COX-2 inhibitors is attributed to their ability to reduce PGI_2_ and increase TXA_2_ biosynthesis in vascular walls and platelets [154]. A disrupted production balance in favor of TXA_2_ over PGI_2_, the most important vascular protector, is shown to be responsible, at least in part, for the pro-thrombotic and pro-atherogenic effects [155]. A clinical trial of a small group of ischemic stroke patients confirmed that treatment with intravenous prostacyclin infusions results in clinical improvement, with regression of hemiplegia or hemiparesis, the disappearance of aphasia, and clearing of consciousness within a few hours after administration [156]. Vasodilation of cerebral microvessels is the effect that may account for the benefit of PGI_2_ on post-ischemic brain tissue. Interestingly, statins, apart from their lipid-lowering activity, have also been shown to be effective in reducing TXA_2_ levels in ischemic stroke patients [157]; thus, extensive studies should investigate the role of statin use as a post-stroke treatment.

The lowering of leukotriene B_4_ synthesis as an ischemic stroke treatment has also been intensively investigated. For example, the experimental inhibition of 5-lipoxygenase with zileuton, an anti-asthmatic drug, attenuates inflammation in the ischemic zone, reduces brain damage, neuronal apoptosis, infarct volume and even improves neurological deficits [133,134,135]. Similar neuroprotection against the injury caused by ischemia-reperfusion is also mediated by the regulation of 5-lipoxygenase expression by microRNA [158]. The disruption or inhibition of the FLAP to block leukotriene synthesis is as efficient in reducing brain edema and neuroinflammation [159]. Antagonists of cysteinyl-leukotriene receptor 1, pranlukast [136], or receptor 2, HAMI 3379 [160], protect against cerebral ischemic injury, ameliorate neuron loss, inhibit astrocyte proliferation, decrease cytokines release, microglial activation and neutrophil accumulation in ischemic regions.

The level of 20-HETE can be effectively modulated by selective inhibitors, such as TS-011 or HET0016, or by the 20-HETE antagonist, 20-hydroxyeicosa-6(Z),15(Z)-dienoic acid. Administration of these agents results in reduced infarct volume, improved microcirculation in the infarct area, and better post-ischemic neurological outcomes in animal stroke models [147,161,162,163]. However, as the role of 20-HETE is reported to be both detrimental and beneficial for stroke pathology, the appropriate modulation of the level of this eicosanoid may be challenging in diverse post-stroke stages.

Increasing lipoxin A_4_ levels seem to be a promising treatment approach; however, physiologically, LXA_4_ is rapidly inactivated. Thus, synthetic analogues could be more promising agents for that kind of stroke therapy. For example, the BML-111 analog of LXA_4_ reduces the infarct volume and improves sensorimotor function in the early post-ischemic phase in rats, but does not affect behavioral deficits in the long-term [164]. However, the administration of an LXA_4_ analog was efficient in reducing levels of pro-inflammatory cytokines and chemokines and increasing the anti-inflammatory cell populations in the post-ischemic brain. Another lipoxin A_4_ analog, LXA_4_ methyl ester (LXA_4_ME), was shown to reduce the brain injury by ameliorating the blood–brain barrier dysfunction in a rat model of transient or permanent cerebral ischemic injury [165].

Reduction of the infarct volume, apoptosis in the ischemic area and the amelioration of neurological deficits can also be achieved by blocking of the downstream EET metabolism by inhibitors of sEH [166,167,168]. Both cell culture and animal studies prove that sEH inhibitors mediate cerebral protection by increasing the levels of EETs. For example, astrocytes treated with sEH inhibitors increase the concentration of EETs after the ischemia-like event and, as a result, release higher levels of protective neurotrophic factors, such as vascular endothelial growth factor (VEGF), which prevent neuronal cell death [167]. Another study showed that 14,15-EET itself or the inhibition of sEH attenuates neuronal apoptosis, astrogliosis and microglia activation, reduces inflammatory responses and promotes angiogenesis in the rat brain after occlusion of the middle cerebral artery [169].

## 5. Conclusions

Stroke is a devastating brain injury with tremendous consequences for human health. Understanding the pathophysiology of ischemic stroke is critical for reducing the burden of the disease or developing therapies. Lipids and lipid mediators, such as eicosanoids, largely contribute to the pathophysiology of ischemic stroke and possible mechanisms of their involvement, discussed in this review, are summarized on Figure 3.

The prevalence of stroke continuously increases because the elderly population is growing faster than the populations of other ages. Stroke is one of the main causes leading to disability and reduced quality of life worldwide; thus, the development of preventive and therapeutic interventions is urgently needed. Approved treatments for ischemic stroke are limited to aspirin, recombinant tissue plasminogen activator (rtPA) and mechanical thrombectomy. Each year, we observe the progress in stroke research, but the findings are not always successfully translated into the clinic.

An effective stroke treatment must focus on rescue of the penumbra zone from ischemic injury to preserve the viability and vitality of as many cells in this zone as possible. Studies on cell culture and animal stroke models have indicated many potential targets for therapeutic interventions within the biosynthetic pathways of lipids and lipid mediators, especially eicosanoids, derived from polyunsaturated fatty acids (summarized in Table 1). Specific enzyme inhibitors, expression modulators or receptor antagonists are able to redirect the metabolite flow in a way that beneficial rather than detrimental effects in the post-stoke brain tissue can be obtained. Neuroprotective, anti-inflammatory and pro-angiogenic effects of these experimental treatments attenuate ischemic brain damage, reduce the infarct area, boost the microcirculation and improve neurological deficits and recovery; however, a lot of research is necessary to determine the molecular mechanisms, efficacy and safety of that kind of intervention.

A number of serum lipids are associated with the risk of ischemic stroke. In clinical practice, the most common laboratory tests determine only the traditional lipid profile. However, research findings show that, for example, lipid ratios are better predictors of vascular risk than a single isolated lipid value. Conventional serum lipid measurements often ignore additional parameters like composition, particle size or the concentration of lipid subclass. Studies show that non-traditional lipids may also be good predictors of stroke risk. More evidence is needed to fully unravel the relationship between lipids, type of lipids, lipid profile (traditional and non-traditional) and stroke. However, it is worth considering including non-traditional lipid measures in daily clinical practice to better predict the risk of stroke in a patient or monitor the recovery of a patient who had already had a stroke episode.

## Figures and Tables

**Figure 1 ijms-21-03618-f001:**
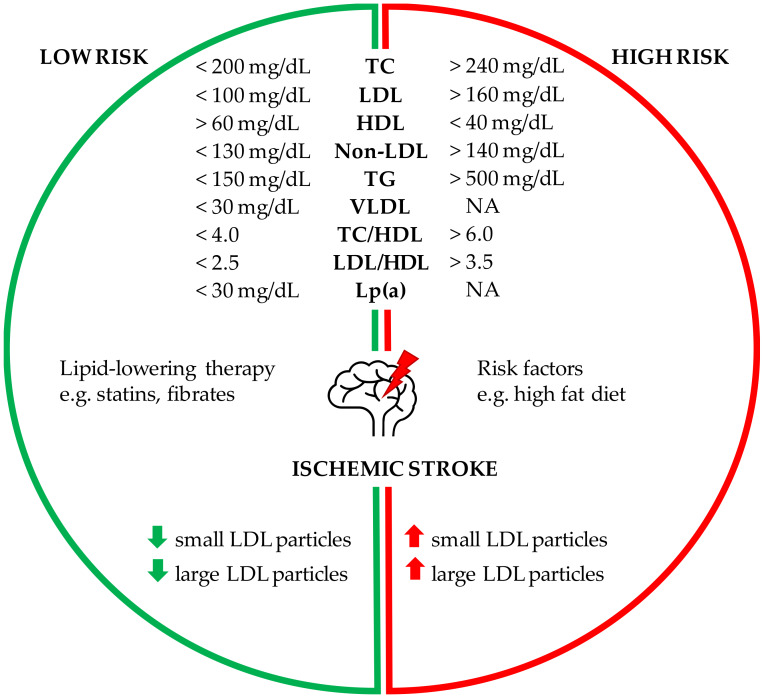
Lipid profile components and the risk of ischemic stroke. The reference values of the specified parameters may vary slightly according to different diagnostic recommendations. The arrow down indicates a low number of particles; the arrow up indicates a high number of particles. NA—not analyzed.

**Figure 2 ijms-21-03618-f002:**
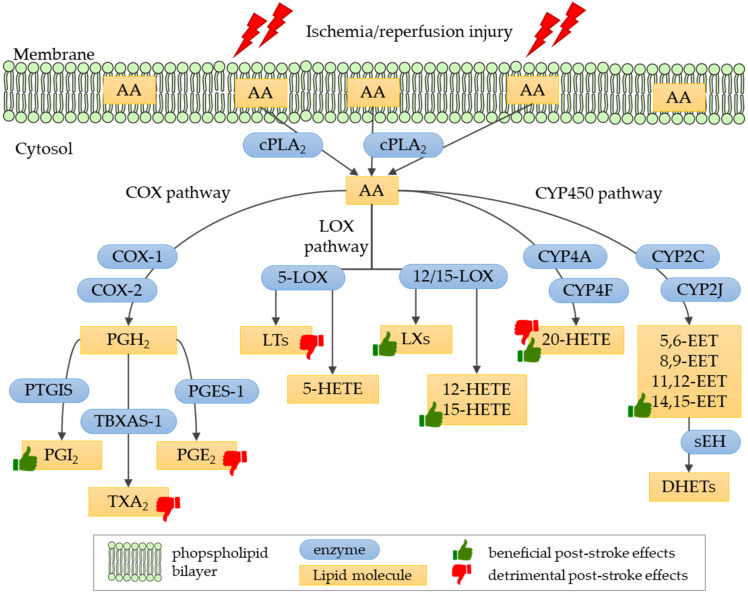
Eicosanoid biosynthesis pathways. Enzymes involved in biosynthetic pathways are denoted as: cPLA_2_, cytosolic phospholipase A_2_; COX-1, cyclooxygenase 1; COX-2, cyclooxygenase 2; PGIS, prostaglandin-I synthase; TBXAS-1, thromboxane-A synthase; PGES-1, prostaglandin E synthase; 12/15-LOX, 12/15-lipoxygenase; 5-LOX, 5-lipoxygenase; CYP4A, cytochrome P450 4A subfamily; CYP4F, cytochrome P450 4F subfamily; CYP2C, cytochrome P450 2C subfamily; CYP2J, cytochrome P450 2J subfamily; sEH, soluble epoxide hydrolase. Lipid molecules generated in biosynthesis are denoted as: AA, arachidonic acid; PGH_2_, prostaglandin H_2_; PGI_2_, prostaglandin I_2_, prostacyclin; TXA_2_, thromboxane A_2_; PGE_2_, prostaglandin E_2_; LTs, leukotrienes; 5-HETE, 5-hydroxyeicosatetraenoic acid; LXs, lipoxins; 12-HETE, 12-hydroxyeicosatetraenoic acid; 15-HETE, 15-hydroxyeicosatetraenoic acid; 20-HETE, 20-hydroxyeicosatetraenoic acid; 5,6-EET, 5,6-epoxyeicosatrienoic acid; 8,9-EET, 8,9-epoxyeicosatrienoic acid; 11,12-EET, 11,12-epoxyeicosatrienoic acid; 14,15-EET, 14,15-epoxyeicosatrienoic acid; DHETs, dihydroxyeicosatrienoic acids. Thumb symbol denotes beneficial or detrimental effects on the post-stroke brain of eicosanoids discussed in this review.

**Figure 3 ijms-21-03618-f003:**
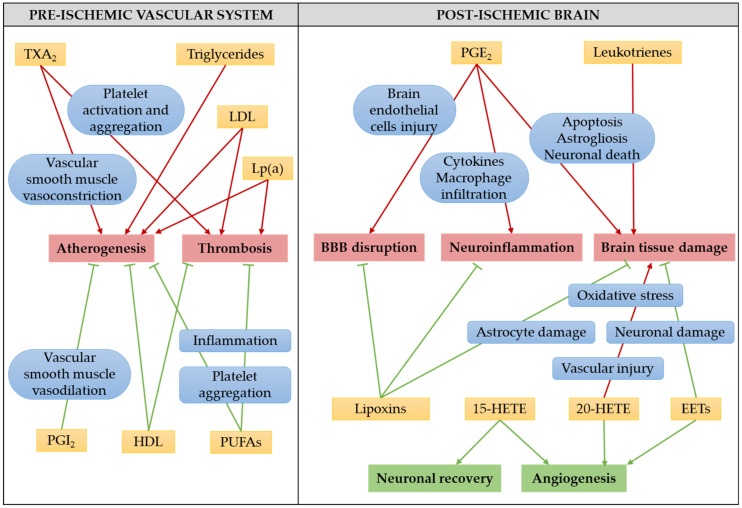
Possible mechanisms of involvement of serum lipids and eicosanoids in the pathophysiology of ischemic stroke. Arrows indicate stimulation, while bar-headed lines indicate inhibition. Green color stands for beneficial, while red for detrimental effects. Abbreviations used: 15-HETE, 15-hydroxyeicosatetraenoic acid; 20-HETE, 20-hydroxyeicosatetraenoic acid; EETs, epoxyeicosatrienoic acids; HDL, high-density lipoprotein; LDL, low-density lipoprotein; Lp(a), lipoprotein(a); PGE_2_, prostaglandin E_2_; PGI_2_, prostaglandin I_2_, prostacyclin; PUFAs, polyunsaturated fatty acids; TXA_2_, thromboxane A_2_.

**Table 1 ijms-21-03618-t001:** Therapeutic interventions targeting lipids and lipid mediators in the prevention and treatment of ischemic stroke.

Therapeutic Intervention ^1^	Effect	Application ^2^	Reference
*Lipid-lowering therapies for stroke prevention*			
Lowering cholesterol with statins	Anti-atherogenic: LDL cholesterol decrease	Clinical use	[91,92,97]
Lowering cholesterol with PCSK9 inhibitors, siRNAs, mimetic peptides, adnectins, vaccinations	Anti-atherogenic: increased LDL-C clearance from circulation	Experimental	[99,100,101,102,103,104,105]
Reduction of hypertriglyceridemia with fibric acid and derivatives	Anti-atherogenic: triglyceride decrease; HDL cholesterol increase	Clinical use	[106,107]
Lowering Lp(a) levels with niacin, PCSK9 or CETP inhibitors, or antisense oligonucleotides	Anti-atherogenic: reduced Lp(a) in circulation	Experimental	[108,109]
*Eicosanoid-targeted therapies for post-ischemic brain tissue rescue*			
Lowering prostaglandins level with COX inhibitors	Anti-atherogenic: anti-platelet effect of aspirin Neuroprotection: anti-inflammatory actions	Clinical use Experimental	[152,153,170]
Increasing prostacyclin levels with intravenous infusions	Vasodilation of cerebral microvessels	Experimental	[156]
Lowering TXA_2_ levels with statins	Anti-thrombogenic: TXA_2_ level decrease, anti-platelet activity	Experimental	[157]
Decreasing leukotriene B_4_ synthesis with 5-LOX inhibitors, microRNAs, FLAP inhibitors, or CysLT receptors antagonists	Neuroprotection: anti-inflammatory and anti-apoptotic actions, reduced neuronal loss	Experimental	[133,134,135,136,158,159,160]
Increasing lipoxin A_4_ levels with LXA_4_ or LXA_4_ME analogues	Neuroprotection: anti-inflammatory action, blood–brain barrier rescue	Experimental	[164,165]
Lowering 20-HETE levels with inhibitors or antagonists	Neuroprotection: improved microcirculation	Experimental	[147,161,162,163]
Increasing EETs levels with sEH inhibitors	Neuroprotection: anti-apoptotic, anti-inflammatory, pro-angiogenic, astrogliosis-preventive actions	Experimental	[166,167,168,169]

^1^ Abbreviations: 20-HETE, 20-hydroxyeicosatetraenoic acid; 5-LOX, 5-lipoxygenase; CETP, cholesterylester transfer protein; COX, cyclooxygenase; CysLT, cysteinyl leukotriene; EETs, epoxyeicosatrienoic acids; FLAP, 5-LOX-activating protein; Lp(a), lipoprotein(a); LXA_4_, lipoxin A_4_; LXA_4_ME, lipoxin A_4_ methyl ester; PCSK9, proprotein convertase subtilisin/kexin type-9; sEH, soluble epoxide hydrolase; TXA_2_, thromboxane A_2_. ^2^ Clinical use: therapeutic interventions used as a standard prevention or treatment for stroke patients. Experimental: therapeutic interventions tested in preclinical or clinical studies, but currently not translated into the clinic.

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
