# Peer review of "Lipids and Lipid Mediators Associated with the Risk and Pathology of Ischemic Stroke"

_ijms, 2020, doi:10.3390/ijms21103618_

Round 1
Reviewer 1 Report
The review "Lipids and lipid mediators associated with the risk
and pathology of ischemic stroke" represent an overview of different lipids and emphasizes on eicosanoids in the pathology of ischemic.
I have a few suggestions that will help in improving the overall quality of the manuscript.
Please include a table of different clinical and preclinical studies highlighting therapeutic interventions/approaches for lipids/eicosanoids.
Please include a figure - Possible mechanism exhibiting the involvement of lipids and eicosanoids in the pathophysiology of ischemic stroke.
Please highlight the role of metabolic modifiers in ischemic stroke in a separate section
Author Response
#Reviewer 1 Comments and Suggestions for Authors
The review "Lipids and lipid mediators associated with the risk and pathology of ischemic stroke" represent an overview of different lipids and emphasizes on eicosanoids in the pathology of ischemic. I have a few suggestions that will help in improving the overall quality of the manuscript.
> Please include a table of different clinical and preclinical studies highlighting therapeutic interventions/approaches for lipids/eicosanoids.
Answer: In the revised version of our manuscript, we have included a table with the summary of different therapeutic interventions for lipids and eicosanoids in stoke – “Table 1. Therapeutic interventions targeting lipids and lipid mediators in the prevention and treatment of ischemic stroke.” The table describes possible therapeutic interventions, characterizes the effects, indicates if the intervention is of clinical use or is an experimental treatment and gives the relevant references. This table is located on page 16 and referred in the text in Conclusion section (line 645).
> Please include a figure - Possible mechanism exhibiting the involvement of lipids and eicosanoids in the pathophysiology of ischemic stroke.
Answer: In the revised version of our manuscript, we have included a figure that summarizes mechanism of lipids and eicosanoids involvement in the pathophysiology of ischemic stroke – “Figure 3. Possible mechanisms of involvement of serum lipids and eicosanoids in the pathophysiology of ischemic stroke.” This figure is located on page 15 and is referred in the text in line 626.
This new figure in the revised version of the manuscript is preceded with following sentences:
“Stroke is a devastating brain injury with tremendous consequences for human health. Understanding the pathophysiology of ischemic stroke is critical for reducing the burden of the disease or developing therapies. Lipids and lipid mediators, such as eicosanoids, largely contribute to the pathophysiology of ischemic stroke and possible mechanisms of their involvement, discussed in this review, are summarized on Figure 3.”
> Please highlight the role of metabolic modifiers in ischemic stroke in a separate section
Answer: We have included a separate section that summarizes the role of modifiers of eicosanoid biosynthesis in ischemic stroke treatment. The section stands as follows:
“4.3. Eicosanoids as the target for ischemic stroke treatment
The emphasis on the role of eicosanoids in the pathology of ischemic stroke has opened up new perspectives for the development of treatments based on modifications within the biosynthetic pathways of these lipid mediators.
The modulation of prostanoid biosynthesis with cyclooxygenase inhibitors has been proposed as one potential treatment strategy. To date, aspirin is the only antiplatelet agent that is used effectively in the early treatment of acute ischemic stroke. Aspirin irreversibly inhibits COX activity in platelets and prevents the conversion of AA to thromboxane A2. The decline in the risk of mortality and morbidity when aspirin is initiated within 48 hours of acute ischemic stroke is small though significant [152]. Targeting prostanoid biosynthesis with COX-2 inhibitors, such as nonsteroidal anti-inflammatory drugs (NSAIDs), the classical pain medications, or the novel COX-2 selective inhibitors (e.g., celecoxib or rofecoxib), has been shown to reduce edema, neuroinflammation, and infarct size in rodent stroke models [153]. Although administration of these drugs reveals a neuroprotective effect in stroke models, adverse effects, including an increased risk of stroke, occur following long-term usage. A recent increase in the number of heart attacks related the use of COX-2 inhibitors is attributed to their ability to reduce PGI2 and increase TXA2 biosynthesis in vascular walls and platelets [154]. A disrupted production balance in favor of TXA2 over PGI2, the most important vascular protector, is shown to be responsible, at least in part, for the pro-thrombotic and pro-atherogenic effects [155]. A clinical trial of a small group of ischemic stroke patients confirmed that treatment with intravenous prostacyclin infusions results in clinical improvement, with regression of hemiplegia or hemiparesis, the disappearance of aphasia, and clearing of consciousness within a few hours after administration [156]. Vasodilation of cerebral microvessels is the effect that may account for the benefit of PGI2 on post-ischemic brain tissue. Interestingly, statins, apart from their lipid-lowering activity, have also been shown to be effective in reducing TXA2 levels in ischemic stroke patients [157]; thus, extensive studies should investigate the role of statin use as a post-stroke treatment.
The lowering of leukotriene B4 synthesis as an ischemic stroke treatment has also been intensively investigated. For example, the experimental inhibition of 5-lipoxygenase with zileuton, an anti-asthmatic drug, attenuates inflammation in the ischemic zone, reduces brain damage, neuronal apoptosis, infarct volume and even improves neurological deficits [133–135]. Similar neuroprotection against the injury caused by ischemia-reperfusion is also mediated by the regulation of 5-lipoxygenase expression by microRNA [158]. The disruption or inhibition of the FLAP to block leukotriene synthesis is as efficient in reducing brain edema and neuroinflammation [159]. Antagonists of cysteinyl-leukotriene receptor 1, pranlukast [136], or receptor 2, HAMI 3379 [160], protect against cerebral ischemic injury, ameliorate neuron loss, inhibit astrocyte proliferation, decrease cytokines release, microglial activation and neutrophil accumulation in ischemic regions.
The level of 20-HETE can be effectively modulated by selective inhibitors, such as TS-011 or HET0016, or by the 20-HETE antagonist, 20-hydroxyeicosa-6(Z),15(Z)-dienoic acid. Administration of these agents results in reduced infarct volume, improved microcirculation in the infarct area, and better post-ischemic neurological outcomes in animal stroke models [147,161–163]. However, as the role of 20-HETE is reported to be both detrimental and beneficial for stroke pathology, the appropriate modulation of the level of this eicosanoid may be challenging in diverse post-stroke stages.
Increasing lipoxin A4 levels seem to be a promising treatment approach; however, physiologically, LXA4 is rapidly inactivated. Thus, synthetic analogues could be more promising agents for that kind of stroke therapy. For example, the BML-111 analog of LXA4 reduces the infarct volume and improves sensorimotor function in the early post-ischemic phase in rats, but does not affect behavioral deficits in the long-term [164]. However, the administration of an LXA4 analog was efficient in reducing levels of pro-inflammatory cytokines and chemokines and increasing the anti-inflammatory cell populations in the post-ischemic brain. Another lipoxin A4 analog, LXA4 methyl ester (LXA4ME), was shown to reduce the brain injury by ameliorating the blood-brain barrier dysfunction in a rat model of transient or permanent cerebral ischemic injury [165].
Reduction of the infarct volume, apoptosis in the ischemic area and the amelioration of neurological deficits can also be achieved by blocking of the downstream EET metabolism by inhibitors of sEH [166–168]. Both cell culture and animal studies prove that sEH inhibitors mediate cerebral protection by increasing the levels of EETs. For example, astrocytes treated with sEH inhibitors increase the concentration of EETs after the ischemia-like event and, as a result, release higher levels of protective neurotrophic factors, such as vascular endothelial growth factor (VEGF), which prevent neuronal cell death [167]. Another study showed that 14,15-EET itself or the inhibition of sEH attenuates neuronal apoptosis, astrogliosis and microglia activation, reduces inflammatory responses and promotes angiogenesis in the rat brain after occlusion of the middle cerebral artery [169]. “
New references, that were not present in the first version of the manuscript include:
- Bansal S, Sangha KS, Khatri P. Drug treatment of acute ischemic stroke. Am J Cardiovasc Drugs. 2013 Feb;13(1):57–69.
- Ahmad M, Zhang Y, Liu H, Rose ME, Graham SH. Prolonged opportunity for neuroprotection in experimental stroke with selective blockade of cyclooxygenase-2 activity. Brain Res. 2009 Jul;1279:168–73.
Zhao, J., Zhang, X., Dong, L., Wen, Y., & Cui, L. (2014). The many roles of statins in ischemic stroke. Current neuropharmacology, 12(6), 564–574.
Reviewer 2 Report
Stroke is a severe neurological disorder in humans that results from an interruption of the blood . Kloska et al reviewed the relationship between serum lipids and ischemic stroke . A variety of lipid mediators is generated from fatty acids and these molecules may have either neuroprotective or neurodegenerative effects on the post-stroke brain tissue; therefore, they largely contribute to the outcome and recovery from stroke. They also discussed the role of lipid mediators, with particular emphasis on eicosanoids, in the pathology of ischemic stroke. They further analyzed the latest research on potential targets in lipid metabolic pathways for ischemic stroke treatment and on the development of new stroke risk biomarkers for use in clinical practice.
This is a well conducted and interesting review manuscript
I have some comments to do :
1) Epidemiological and clinical trials have shown that n-3 polyunsaturated fatty acids (PUFAs) reduce the incidence of coronary heart disease or stroke. However, the association between PUFAs and acute-phase stroke recently has been thoroughly studied. Authors should add some impact of serum PUFAs on early neurological deterioration (END) in patients with acute ischemic stroke ;
2) Cardioembolic (CE) stroke is the most severe subtype of ischemic stroke with high recurrence and mortality. Recently some studies reported information on the association of plasma fatty acid (FA) with CE stroke. Authors should add some information about the relationship between composition of plasma phospholipid FA and its association with the risk of CE stroke;
3) The effects of n-3 fatty acid supplementation in the form of fresh fish, fish oil, and docosahexaenoic acid (DHA) oil on the fatty acid composition of plasma lipid fractions, and platelets and erythrocyte membranes and on the risk of cerebrovascular events should be treated
4) authors should add some information about the possible therapeutic use of Atorvastatin in the acute phase of stroke and they should add on their reference section these citations :
- Early High-dosage Atorvastatin Treatment Improved Serum Immune-inflammatory Markers and Functional Outcome in Acute Ischemic Strokes Classified as Large Artery Atherosclerotic Stroke: A Randomized Trial.
-
Tuttolomondo A, Di Raimondo D, Pecoraro R, Maida C, Arnao V, Della Corte V, Simonetta I, Corpora F, Di Bona D, Maugeri R, Iacopino DG, Pinto A.
Medicine (Baltimore). 2016 Mar;95(13):e3186. ;
Select item 21677574
The atorvastatin during ischemic stroke study: a pilot randomized controlled trial.
Muscari A, Puddu GM, Santoro N, Serafini C, Cenni A, Rossi V, Zoli M.
Clin Neuropharmacol. 2011 Jul-Aug;34(4):141-7;
Select item 28357403
Effect of Prior Atorvastatin Treatment on the Frequency of Hospital Acquired Pneumonia and Evolution of Biomarkers in Patients with Acute Ischemic Stroke: A Multicenter Prospective Study.
Yu Y, Zhu C, Liu C, Gao Y.
Biomed Res Int. 2017;2017:5642704.;
Select item 2635580
Comparative effects of more versus less aggressive treatment with statins on the long-term outcome of patients with acute ischemic stroke.
Tziomalos K, Giampatzis V, Bouziana SD, Spanou M, Kostaki S, Papadopoulou M, Angelopoulou SM, Konstantara F, Savopoulos C, Hatzitolios AI.
Atherosclerosis. 2015 Nov;243(1):65-70.
Author Response
#Reviewer 2 Comments and Suggestions for Authors
Stroke is a severe neurological disorder in humans that results from an interruption of the blood. Kloska et al reviewed the relationship between serum lipids and ischemic stroke . A variety of lipid mediators is generated from fatty acids and these molecules may have either neuroprotective or neurodegenerative effects on the post-stroke brain tissue; therefore, they largely contribute to the outcome and recovery from stroke. They also discussed the role of lipid mediators, with particular emphasis on eicosanoids, in the pathology of ischemic stroke. They further analyzed the latest research on potential targets in lipid metabolic pathways for ischemic stroke treatment and on the development of new stroke risk biomarkers for use in clinical practice.
This is a well conducted and interesting review manuscript.
I have some comments to do:
> 1) Epidemiological and clinical trials have shown that n-3 polyunsaturated fatty acids (PUFAs) reduce the incidence of coronary heart disease or stroke. However, the association between PUFAs and acute-phase stroke recently has been thoroughly studied. Authors should add some impact of serum PUFAs on early neurological deterioration (END) in patients with acute ischemic stroke ;
Answer: We have added new paragraph with the information on the role of PUFAs in reducing CHD incidents and strokes, and the impact of PUFAs on END.
Page 5, lines 203–244
“2.6. Polyunsaturated fatty acids and risk of ischemic stroke
The n-3 polyunsaturated fatty acids (n-3 PUFAs) are a class of essential unsaturated fatty acids necessary for proper biological activity and function in living organisms. The n-3 PUFAs are poorly synthesised in the human body and have to be orally supplemented. Fish, such as mackerel, salmon, tuna, sardines, herring, and halibut, are a major source of n-3 PUFAs in the human diet, and they contain docosahexaenoic acid (DHA), docosapentaenoic acid (DPA), and eicosapentaenoic acid (EPA) [54]. The n-3 PUFAs have potent anti-inflammatory activity, reduce platelet aggregation, stabilise atherosclerotic plaques, and reduce major cardiovascular risk factors, including hypertension and hyperlipidaemia [55]. They can act as an antioxidant in reducing cerebral lipid peroxides and play a role in regulating oxidative stress by increasing the oxidative burden and improving antioxidant defence capabilities [56]. In addition, n-3 PUFAs trigger other responses, such as neurogenesis and revascularisation in stroke, which could be used in the development of acute-phase ischemic stroke therapy [54]. Moreover, because ischemic stroke is a heterogeneous disorder with different pathophysiological pathways and separate etiological subtypes, various mechanisms of PUFA action should be considered. DHA plays a greater role in reducing the risk of atherothrombotic stroke by reducing endothelial dysfunction and atherosclerosis [57], while EPA and DPA have a greater impact on the risk of cardioembolic stroke because of their effect on clotting and atrial fibrillation [58].
The second key component of a heart-healthy dietary pattern is a high content of n-6 PUFAs, obtained mainly from vegetable oil, nuts, and seeds. Dietary n-6 PUFAs primarily include linoleic acid (LA) and arachidonic acid (AA). The n-6 PUFAs have not generally been associated with stroke risk [59,60].
The Japan EPA Lipid Intervention Study (JELIS) has shown that treatment with highly purified EPA and low-dose statins significantly reduces the incidence of coronary artery diseases and stroke compared to statin therapy alone, without altering the reduction in low-density lipoprotein cholesterol levels [61]. Nishizaki et al. reported that ratios of serum n-3 PUFAs to n-6 PUFAs, such as EPA/AA and DHA/AA ratios, could be useful markers to determine the incidence of coronary events, peripheral artery diseases, and early neurological deterioration after acute ischemic stroke [62]. Thies et al. examined the effect of fish oil administration on plaque regression and found that giving fish oil to patients resulted not only in plaque regression but also an increases in EPA and DHA within the plaque and a decrease in macrophage counts [63]. In addition, Ajami et al. reported that DHA+EPA provided neuroprotection against ischemic brain injury by increasing the levels of antiapoptotic proteins, such as Bcl-2 and Bcl-xL, thereby suppressing the inflammatory response [64].
Attention has also been focused on assessing the relationship between PUFA levels and early neurological deterioration (END) in acute-phase ischemic stroke [65]. END occurs in approximately one-third of patients in the acute phase of ischemic stroke and is associated with neurological and functional decline. It also strongly correlates with poor functional outcome and usually leads to a significant increase in mortality rate [66]. According to a recommended definition, END occurs when there is an increase in the total National Institutes of Health Stroke Scale score of ≥ 2 points within 72 hours 1–3 times a day after admission [67]. Suda et al. revealed that END is negatively associated with the EPA/AA, DHA/AA, and EPA+DHA/AA ratios. The study shows that a low serum n-3 PUFA/n-6 PUFA ratio might be an indication of possible END in patients with acute ischemic stroke, as demonstrated in the population of Japanese stroke patients [65].”
New references in the above text include:
- Bu, J.; Dou, Y.; Tian, X.; Wang, Z.; Chen, G. The Role of Omega-3 Polyunsaturated Fatty Acids in Stroke. 2016.
- Mozaffarian, D.; Wu, J.H.Y. Omega-3 fatty acids and cardiovascular disease: Effects on risk factors, molecular pathways, and clinical events. Am. Coll. Cardiol. 2011, 58, 2047–2067.
- Rebiger, L.; Lenzen, S.; Mehmeti, I. Susceptibility of brown adipocytes to pro-inflammatory cytokine toxicity and reactive oxygen species. Rep. 2016, 36.
- Geleijnse, J.M.; Giltay, E.J.; Grobbee, D.E.; Donders, A.R.T.; Kok, F.J. Blood pressure response to fish oil supplementation: Metaregression analysis of randomized trials. Hypertens. 2002, 20, 1493–1499.
- Mozaffarian, D.; Wu, J.H.Y. (n-3) Fatty Acids and Cardiovascular Health: Are Effects of EPA and DHA Shared or Complementary? Nutr.2012, 142, 614S-625S.
- Wallström, P.; Sonestedt, E.; Hlebowicz, J.; Ericson, U.; Drake, I.; Persson, M.; Gullberg, B.; Hedblad, B.; Wirfält, E. Dietary fiber and saturated fat intake associations with cardiovascular disease differ by sex in the Malmö diet and cancer cohort: A prospective study. PLoS One2012, 7.
- Yaemsiri, S.; Sen, S.; Tinker, L.F.; Robinson, W.R.; Evans, R.W.; Rosamond, W.; Wasserthiel-Smoller, S.; He, K. Serum fatty acids and incidence of ischemic stroke among postmenopausal women. Stroke 2013, 44, 2710–2717.
- Tanaka, K.; Ishikawa, Y.; Yokoyama, M.; Origasa, H.; Matsuzaki, M.; Saito, Y.; Matsuzawa, Y.; Sasaki, J.; Oikawa, S.; Hishida, H.; et al. Reduction in the recurrence of stroke by eicosapentaenoic acid for hypercholesterolemic patients: Subanalysis of the JELIS trial. Stroke 2008, 39, 2052–2058.
- Nishizaki, Y.; Shimada, K.; Tani, S.; Ogawa, T.; Ando, J.; Takahashi, M.; Yamamoto, M.; Shinozaki, T.; Miyauchi, K.; Nagao, K.; et al. Significance of imbalance in the ratio of serum n-3 to n-6 polyunsaturated fatty acids in patients with acute coronary syndrome. J. Cardiol. 2014, 113, 441–445.
- Thies, F.; Garry, J.M.C.; Yaqoob, P.; Rerkasem, K.; Williams, J.; Shearman, C.P.; Gallagher, P.J.; Calder, P.C.; Grimble, R.F. Association of n-3 polyunsaturated fatty acids with stability of atherosclerotic plaques: A randomised controlled trial. Lancet 2003, 361, 477–485.
- Ajami, M.; Eghtesadi, S.; Razaz, J.M.; Kalantari, N.; Habibey, R.; Nilforoushzadeh, M.A.; Zarrindast, M.; Pazoki-Toroudi, H. Expression of Bcl-2 and Bax after hippocampal ischemia in DHA + EPA treated rats. Sci. 2011, 32, 811–818.
- Suda, S.; Katsumata, T.; Okubo, S.; Kanamaru, T.; Suzuki, K.; Watanabe, Y.; Katsura, K.I.; Katayama, Y. Low serum n-3 polyunsaturated fatty acid/n-6 polyunsaturated fatty acid ratio predicts neurological deterioration in japanese patients with acute ischemic stroke. Dis.2013, 36, 388–393.
- Alawneh, J.A.; Moustafa, R.R.; Baron, J.C. Hemodynamic factors and perfusion abnormalities in early neurological deterioration. Stroke 2009, 40.
- Huang, Z.X.; Wang, Q.Z.; Dai, Y.Y.; Lu, H.K.; Liang, X.Y.; Hu, H.; Liu, X.T. Early neurological deterioration in acute ischemic stroke: A propensity score analysis. Chinese Med. Assoc. 2018, 81, 865–870.
> 2) Cardioembolic (CE) stroke is the most severe subtype of ischemic stroke with high recurrence and mortality. Recently some studies reported information on the association of plasma fatty acid (FA) with CE stroke. Authors should add some information about the relationship between composition of plasma phospholipid FA and its association with the risk of CE stroke;
Answer: We have added additional information about the relationship between composition of FA and its association with the risk of cardioembolic stroke with corresponding references:
Page 6, lines 245–273
“2.7. Fatty acids and cardioembolic stroke
Elevated fatty acid (FA) levels are associated with several risk factors for atherosclerosis, including abdominal obesity [68], arterial hypertension [69], and insulin resistance [70], as well as coronary artery disease (CAD) [71], arrhythmia [72], and atrial fibrillation [73]. Because an association with both atherosclerosis and arrhythmia has been reported, FAs may correlate with ischemic stroke. However, the effect of FAs on ischemic stroke is poorly understood. Because ischemic stroke is a heterogeneous disorder with a variety of pathophysiological pathways, including atherothrombosis and cardioembolism, the etiological subtypes of ischemic stroke should be analyzed separately [74]. Almost all long-term correlation studies of FAs and stroke risk estimate the level of FAs obtained with food on the basis of self-reported questionnaires. Unfortunately, this is unclear for individual FAs that are not well separated by dietary questionnaire data. For a more detailed assessment, Saber et al. measured the levels of circulating n-3 PUFA phospholipids and examined their association with ischemic stroke incidence, including atherosclerotic and cardioembolic stroke subtypes [75]. Patients evaluated for phospholipid levels were recruited for three separate prospective cohort studies in the US: the Cardiovascular Health Study (CHS), Nurses’ Health Study (NHS), and Health Professionals Follow-Up Study (HPFS) [75]. The authors of these studies show that among ischemic stroke subtypes, DHA is inversely associated with atherothrombotic stroke and DPA is associated with cardioembolic stroke. These relationships remain significant after including demographic, lifestyle, and vascular risk factors. By comparison, EPA is not associated with total ischemic, atherothrombotic, or cardioembolic stroke. The authors confirmed the hypothesis that individual FAs in serum have various associations with ischemic, atherothrombotic, and cardioembolic stroke [75]. Earlier studies have shown that elevated FA levels are associated with cardioembolic (CE) stroke, but this association was not seen in non-CE stroke. Atrial fibrillation may potentially act as an intermediary between FAs and stroke caused by cardioembolism [73]. Another study also found that an elevated FA concentration may serve as a marker of stroke caused by cardioembolism. In addition, the assessment of FA concentration can predict the stroke recurrence following a CE stroke [76]. In patients with acute stroke, significantly elevated FA levels are observed in groups with a higher risk of cardioembolism. These results suggest that enhanced thrombogenicity may be the main mechanism explaining elevated FA levels in patients with cardioembolic stroke [77].”
New references in the above text include:
- Poirier, P.; Giles, T.D.; Bray, G.A.; Hong, Y.; Stern, J.S.; Pi-Sunyer, F.X.; Eckel, R.H. Obesity and cardiovascular disease: Pathophysiology, evaluation, and effect of weight loss: An update of the 1997 American Heart Association Scientific Statement on obesity and heart disease from the Obesity Committee of the Council on Nutrition, Physical Activity, and Metabolism. Circulation 2006, 113, 898–918.
- Fagot-Campagna, A.; Balkau, B.; Simon, D.; Warnet, J.-M.; Claude, J.-R.; Ducimetiere, P.; Eschwege, E. High free fatty acid concentration: an independent risk factor for hypertension in the Paris Prospective Study; 1998; Vol. 27;.
- Bays, H.; Mandarino, L.; DeFronzo, R.A. Role of the Adipocyte, Free Fatty Acids, and Ectopic Fat in Pathogenesis of Type 2 Diabetes Mellitus: Peroxisomal Proliferator-Activated Receptor Agonists Provide a Rational Therapeutic Approach. Clin. Endocrinol. Metab. 2004, 89, 463–478.
- O’Donoghue, M.; De Lemos, J.A.; Morrow, D.A.; Murphy, S.A.; Buros, J.L.; Cannon, C.P.; Sabatine, M.S. Prognostic utility of heart-type fatty acid binding protein in patients with acute coronary syndromes. Circulation 2006, 114, 550–557.
- Cocco, G.; Chu, D. Drug points: Rimonabant may induce atrial fibrillation. BMJ 2009, 339, 296.
- Khawaja, O.; Bartz, T.M.; Ix, J.H.; Heckbert, S.R.; Kizer, J.R.; Zieman, S.J.; Mukamal, K.J.; Tracy, R.P.; Siscovick, D.S.; Djoussé, L. Plasma free fatty acids and risk of atrial fibrillation (from the Cardiovascular Health Study). In Proceedings of the American Journal of Cardiology; NIH Public Access, 2012; Vol. 110, pp. 212–216.
- Iso, H.; Sato, S.; Umemura, U.; Kudo, M.; Koike, K.; Kitamura, A.; Imano, H.; Okamura, T.; Naito, Y.; Shimamoto, T. Linoleic acid, other fatty acids, and the risk of stroke. Stroke 2002, 33, 2086–2093.
- Saber, H.; Yakoob, M.Y.; Shi, P.; Longstreth, W.T.; Lemaitre, R.N.; Siscovick, D.; Rexrode, K.M.; Willett, W.C.; Mozaffarian, D. Omega-3 fatty acids and incident ischemic stroke and its atherothrombotic and cardioembolic subtypes in 3 US cohorts.
- Choi, J.Y.; Kim, J.S.; Kim, J.H.; Oh, K.; Koh, S.B.; Seo, W.K. High free fatty acid level is associated with recurrent stroke in cardioembolic stroke patients. Neurology 2014, 82, 1142–1148.
- Seo, W.K.; Jung, J.M.; Kim, J.H.; Koh, S.B.; Bang, O.Y.; Oh, K. Free fatty acid is associated with thrombogenicity in cardioembolic stroke. Dis. 2017, 44, 160–168.
> 3) The effects of n-3 fatty acid supplementation in the form of fresh fish, fish oil, and docosahexaenoic acid (DHA) oil on the fatty acid composition of plasma lipid fractions, and platelets and erythrocyte membranes and on the risk of cerebrovascular events should be treated
Answer: We added the information and references about the influence of diet-derived FA on composition of plasma lipid fraction, and platelets and erythrocyte membranes and on the risk of cerebrovascular events.
Page 7, lines 274–298
“2.8. The role of fish-derived fatty acids in stroke
For many years, intensive research has been carried out to assess the biological effects of consuming fish-derived PUFAs [78]. These studies confirm the view that n-3 PUFAs can affect several cellular processes known to be important in the development of cardiovascular disease, stroke, and protective effects [79],[80]. Most studies have used doses of fish oils exceeding what is usually found in the diet. Surprisingly, significant vascular benefits are observed even with modest fish consumption [81]. Long-term studies have shown that increased intake of n-3 PUFAs, in particular EPA and DHA, can have a beneficial effect on serum lipids [82], platelet aggregation [83], and bleeding time [84] and, thus, may lead to a reduced risk of atherosclerosis and thrombotic complications [63]. Prolonged fish consumption leads to increased incorporation of n-3 PUFAs into plasma lipids, erythrocytes, and platelets [85]. After supplementation with fish oil, elevated EPA and DHA content in plasma lipids, platelets, and erythrocyte membranes are observed with a simultaneous decrease in AA content [86]. In order to compare the effects of n-3 PUFA consumption, the following four fish-derived sources of n-3 PUFA were used: three rich sources (raw fatty fish [smoked salmon], cooked fatty fish [salmon fillet], or fish oil [cod liver oil]) and one poor source (fish low in n-3 PUFA [cod fillet]). Therefore, the following blood parameters were assessed: blood lipid composition and functional properties of blood cells, as measured by the potential of lipopolysaccharide (LPS) to generate activation products in whole blood [87]. Elvevoll et al. did not notice any significant differences between the effects of eating cooked fish compared to raw fish (smoked salmon). They found that the intake of fatty fish is more effective in increasing EPA and DHA than supplementing with fish oil and is more likely to have a beneficial effects on HDL cholesterol and whole blood activation reactions [87]. However, another group of researchers showed that the action of DHA-rich oil (without EPA) had a comparable hypotriglyceridaemic effect as a fish diet and fish oil supplementation. Moreover, a fish diet and fish oil supplementation increased the proportion of n-3 PUFA in plasma lipids, platelets, and erythrocyte membranes [88].”
New references in the above text include:
- Erkkilä, A.T.; Lehto, S.; Pyörälä, K.; Uusitupa, M.I.J. n-3 Fatty acids and 5-y risks of death and cardiovascular disease events in patients with coronary artery disease. J. Clin. Nutr. 2003, 78, 65–71.
- Watanabe, Y.; Tatsuno, I. Omega-3 polyunsaturated fatty acids for cardiovascular diseases: present, past and future. Expert Rev. Clin. Pharmacol. 2017, 10, 865–873.
- Owen, A.J.; Magliano, D.J.; O’Dea, K.; Barr, E.L.M.; Shaw, J.E. Polyunsaturated fatty acid intake and risk of cardiovascular mortality in a low fish-consuming population: a prospective cohort analysis. J. Nutr. 2016, 55, 1605–13.
- Wennberg, M.; Jansson, J.-H.; Norberg, M.; Skerfving, S.; Strömberg, U.; Wiklund, P.-G.; Bergdahl, I.A. Fish consumption and risk of stroke: a second prospective case-control study from northern Sweden. J. 2016, 15, 98.
- Mensink, R.P. Effects of saturated fatty acids on serum lipids an lipoproteins: a systematic review and regression analysis;
- McEwen, B.J.; Morel-Kopp, M.C.; Chen, W.; Tofler, G.H.; Ward, C.M. Effects of omega-3 polyunsaturated fatty acids on platelet function in healthy subjects and subjects with cardiovascular disease. Thromb. Hemost. 2013, 39, 25–32.
- Cohen, M.G.; Rossi, J.S.; Garbarino, J.; Bowling, R.; Motsinger-Reif, A.A.; Schuler, C.; Dupont, A.G.; Gabriel, D. Insights into the inhibition of platelet activation by omega-3 polyunsaturated fatty acids: Beyond aspirin and clopidogrel. Res. 2011, 128, 335–340.
- Browning, L.M.; Walker, C.G.; Mander, A.P.; West, A.L.; Madden, J.; Gambell, J.M.; Young, S.; Wang, L.; Jebb, S.A.; Calder, P.C. Incorporation of eicosapentaenoic and docosahexaenoic acids into lipid pools when given as supplements providing doses equivalent to typical intakes of oily fish. J. Clin. Nutr. 2012, 96, 748–758.
- Ghasemi Fard, S.; Wang, F.; Sinclair, A.J.; Elliott, G.; Turchini, G.M. How does high DHA fish oil affect health? A systematic review of evidence. Rev. Food Sci. Nutr. 2019, 59, 1684–1727.
- Elvevoll, E.O.; Barstad, H.; Breimo, E.S.; Brox, J.; Eilertsen, K.E.; Lund, T.; Olsen, J.O.; Østerud, B. Enhanced incorporation of n-3 fatty acids from fish compared with fish oils. Lipids 2006, 41, 1109–1114.
- Vidgren, H.M.; Ågren, J.J.; Schwab, U.; Rissanen, T.; Hänninen, O.; Uusitupa, M.I.J. Incorporation of n-3 fatty acids into plasma lipid fractions, and erythrocyte membranes and platelets during dietary supplementation with fish, fish oil, and docosahexaenoic acid-rich oil among healthy young men. Lipids 1997, 32, 697–705.
> 4) authors should add some information about the possible therapeutic use of Atorvastatin in the acute phase of stroke and they should add on their reference section these citations :
- Early High-dosage Atorvastatin Treatment Improved Serum Immune-inflammatory Markers and Functional Outcome in Acute Ischemic Strokes Classified as Large Artery Atherosclerotic Stroke: A Randomized Trial.
- Tuttolomondo A, Di Raimondo D, Pecoraro R, Maida C, Arnao V, Della Corte V, Simonetta I, Corpora F, Di Bona D, Maugeri R, Iacopino DG, Pinto A. Medicine (Baltimore). 2016 Mar;95(13):e3186. ; Select item 21677574
- The atorvastatin during ischemic stroke study: a pilot randomized controlled trial. Muscari A, Puddu GM, Santoro N, Serafini C, Cenni A, Rossi V, Zoli M. Clin Neuropharmacol. 2011 Jul-Aug;34(4):141-7; Select item 28357403
- Effect of Prior Atorvastatin Treatment on the Frequency of Hospital Acquired Pneumonia and Evolution of Biomarkers in Patients with Acute Ischemic Stroke: A Multicenter Prospective Study. Yu Y, Zhu C, Liu C, Gao Y. Biomed Res Int. 2017;2017:5642704.; Select item 2635580
- Comparative effects of more versus less aggressive treatment with statins on the long-term outcome of patients with acute ischemic stroke. Tziomalos K, Giampatzis V, Bouziana SD, Spanou M, Kostaki S, Papadopoulou M, Angelopoulou SM, Konstantara F, Savopoulos C, Hatzitolios AI. Atherosclerosis. 2015 Nov;243(1):65-70.
Answer: We have included the information on atorvastatin therapy in section 2.9. and have citied the indicated references.
Page 8, lines 305–320
“Moreover, more aggressive statin treatment improves the long-term functional outcome of patients discharged after an acute ischemic stroke more than less aggressive treatment [93]. One of the most commonly used statins to lower high cholesterol is atorvastatin. It is a common statin used to investigate the efficacy of statin therapy, especially high-intensity statin therapy in patients with ischemic stroke. Preliminary studies have shown that in patients with recent stroke or transient ischemic attack and without coronary heart disease, 80 mg atorvastatin daily reduces the overall incidence of stroke and cardiovascular events, despite a slight increase in the incidence of hemorrhagic stroke [94]. Many studies have confirmed the effect of atorvastatin at lowering lipids and decreasing the number and frequency of vascular incidents, as well as its clinical efficacy at reducing the burden of disease after stroke or transient ischemic attack [95],[96]. Atorvastatin improves endothelial function, enhances the stability of atherosclerotic plaque, and inhibits inflammatory and thrombogenic responses in arterial walls [97]. In the latest Stroke Prevention by Aggressive Reduction in Cholesterol Levels (SPARCLs) cohort trial, atorvastatin was compared with a placebo in patients with recent stroke or transient ischemic attack. Atorvastatin was found to reduce the first occurrence of stroke and the first occurrence of composite vascular events [98].”
New references in the above text include:
- Tziomalos, K.; Giampatzis, V.; Bouziana, S.D.; Spanou, M.; Kostaki, S.; Papadopoulou, M.; Angelopoulou, S.M.; Konstantara, F.; Savopoulos, C.; Hatzitolios, A.I. Comparative effects of more versus less aggressive treatment with statins on the long-term outcome of patients with acute ischemic stroke. Atherosclerosis 2015, 243, 65–70.
- Amarenco, P.; Bogousslavsky, J.; Callahan, A.; Goldstein, L.B.; Hennerici, M.; Rudolph, A.E.; Sillesen, H.; Simunovic, L.; Szarek, M.; Welch, K.M.A.; et al. High-dose atorvastatin after stroke or transient ischemic attack. N. Engl. J. Med. 2006, 355, 549–559.
- Muscari, A.; Puddu, G.M.; Santoro, N.; Serafini, C.; Cenni, A.; Rossi, V.; Zoli, M. The atorvastatin during ischemic stroke study: A pilot randomized controlled trial. Clin. Neuropharmacol. 2011, 34, 141–147.
- Tuttolomondo, A.; Di Raimondo, D.; Pecoraro, R.; Maida, C.; Arnao, V.; Corte, V. Della; Simonetta, I.; Corpora, F.; Di Bona, D.; Maugeri, R.; et al. Early high-dosage atorvastatin treatment improved serum immune-inflammatory markers and functional outcome in acute ischemic strokes classified as large artery atherosclerotic stroke: A randomized trial. Med. (United States) 2016, 95, e3186.
- Yu, Y.; Zhu, C.; Liu, C.; Gao, Y. Clinical Study Effect of Prior Atorvastatin Treatment on the Frequency of Hospital Acquired Pneumonia and Evolution of Biomarkers in Patients with Acute Ischemic Stroke: A Multicenter Prospective Study. 2017.
- Szarek, M.; Amarenco, P.; Callahan, A.; DeMicco, D.; Fayyad, R.; Goldstein, L.B.; Laskey, R.; Sillesen, H.; Welch, K.M. Atorvastatin Reduces First and Subsequent Vascular Events Across Vascular Territories: The SPARCL Trial. Am. Coll. Cardiol. 2020, 75, 2110–2118.
Reviewer 3 Report
The paper written by the following Authors from Poland: Anna Kloska, Marcelina Malinowska, Magdalena Gabig-Ciminska and Joanna Jakobkiewicz-Banecka, entitled “Lipids and lipid mediators associated with the risk 3 and pathology of ischemic stroke”, presents an informative review in the topic of lipids and lipid mediators associated with ischemic stroke. It discusses the role of serum lipids and lipid mediators, eicosanoids, in stroke development. The authors also briefly describe the latest targets in lipid metabolic pathways potentially beneficial in ischemic stroke.
Author Response
#Reviewer 3 Comments and Suggestions for Authors
The paper written by the following Authors from Poland: Anna Kloska, Marcelina Malinowska, Magdalena Gabig-Ciminska and Joanna Jakobkiewicz-Banecka, entitled “Lipids and lipid mediators associated with the risk 3 and pathology of ischemic stroke”, presents an informative review in the topic of lipids and lipid mediators associated with ischemic stroke. It discusses the role of serum lipids and lipid mediators, eicosanoids, in stroke development. The authors also briefly describe the latest targets in lipid metabolic pathways potentially beneficial in ischemic stroke.
Answer: No revisions were asked by the reviewer.